# Feasibility of a Planar Coil-Based Inductive-Capacitive Water Level Sensor with a Quality-Detection Feature: An Experimental Study

**DOI:** 10.3390/s22155508

**Published:** 2022-07-23

**Authors:** Lakshmi Areekath, Gaurav Lodha, Subham Kumar Sahana, Boby George, Ligy Philip, Subhas Chandra Mukhopadhyay

**Affiliations:** 1Department of Electrical Engineering, IIT Madras, Chennai 600036, India; lareekath1721@gmail.com (L.A.); boby@ee.iitm.ac.in (B.G.); 2Department of Civil Engineering, IIT Madras, Chennai 600036, India; glodha0@gmail.com (G.L.); subhamkumarsahana2@gmail.com (S.K.S.); ligy@iitm.ac.in (L.P.); 3Mechanical/Electronics Engineering, School of Engineering, Macquarie University, Sydney, NSW 2109, Australia

**Keywords:** level sensor, inductive sensor, capacitive sensor, eddy current sensing, conductivity of water, water quality

## Abstract

This paper presents a new water-level-sensing mechanism based on planar coils fabricated on a printed circuit board (PCB). In addition to level, the sensor detects any relative increase in conductivity compared to that of clean water, which is an indicator of its quality. The sensing mechanism utilizes the eddy current induced in the water column, the corresponding change in the coil inductance, and the change in the turn-to-turn capacitance of the coil in the presence of water. Although several level sensors are available, there is none that gives the level and quality information using a single sensing element. Since both water quantity and quality measurements are fundamental in realizing efficient water and wastewater management, obtaining these two parameters from the same sensor is very beneficial. A scalable, planar coil-based sensor that helps achieve this goal is designed, fabricated, and tested in a laboratory setting. The results illustrate that the reactance of the sensor coil measured at a frequency (1 kHz for the prototype) much lower than the self-resonance of the coil gives reliable information about the level of water, while the measurement made at resonance, using an inductance-to-digital converter, is a clear indicator of its conductivity and, hence, quality.

## 1. Introduction

Access to clean water for all, through its sustainable management, is ranked sixth among the United Nations’ seventeen goals [1] for Sustainable Development. If studied carefully, most of the goals listed, ranging from food security to combating climate change, rely on efficient water handling. As we know, water is a crucial element in almost every sector. For instance, consider the agricultural sector. Its productivity is heavily contingent upon water availability in sufficient quantity and quality. Although Earth is a planet with 71% of its composition being water, about 97% of the water is in the form of saline water. Hence the needs of all, barring the marine organisms, have to be met by the remaining 3%. A large portion of this freshwater is frozen in glaciers, ice caps, or lies deep underground in aquifers. This leaves about 1% of freshwater accessible to meet the needs of every living thing on Earth [2]. While climate change and increasing demand remain huge challenges that need long-term intervention, it is important that the loopholes such as wastage of water and its inefficient usage [3], prevalent in the present water-management system, be sealed as soon as possible. 

In the past, monitoring widespread water distribution networks for drinking water or irrigation or waste water management was not easy. Currently, this situation is much improved, owing to the advent of the Internet-of-Things (IoT) and established technologies of cloud computing and artificial intelligence [4,5]. Though IoT can help to realize a smart network, the availability of reliable sensor data indicating the quantity and quality of water is a key component. 

The quantity of water in a storage unit, e.g., a tank, is determined through level measurement. Level measurement is broadly classified into (i) point-level, e.g., float switches, vibrating forks that give an output when the level crosses a preset value, and (ii) continuous-level, where differential pressure, radar, image, ultrasonic, optical, capacitive, or inductive sensors are used to provide dynamic level measurement [6,7,8,9]. Though engineers traditionally have used point-level sensors in water and wastewater utilities, the continuous-level devices are much more helpful in managing the overall system efficiently [4].

Real-time, continuous level measurement is imperative for smart water management infrastructure. The commonly used differential pressure-based level measurement [10] is affected by mounting constraints and specific gravity/density changes. It is also costly as periodic calibration and maintenance are required to ensure the required accuracy. Similar is the case with radar and ultrasonic-type measurement methods. These sensors need to be installed on the lid of the tank/storage unit. Additionally, the unwanted reflections caused by barriers on the liquid surface affect their functioning, such as foam and surface ripples [11,12]. Typically, for continuous optical fiber liquid-level sensing, the most popular class is the pressure-sensitive detector, including the microbend [13], Bragg grating [14], and Fabry–Perot [15,16]. These systems are relatively expensive and need regular maintenance. Capacitive-type liquid level measurement systems need no moving parts, have low power consumption, and exhibit linearity [17]. If measuring a nonconductive material, that material is used as the insulator part of the capacitor and the conductive tank wall acts as one of the conductor plates with the immersed electrode acting as the other [18]. 

In general, inductive-based sensors [17] have several advantages, such as no requirement of direct contact with the medium of interest, low cost, less maintenance, and measurement systems required are simple [19,20] less sensitive to dust, oil etc. However, their use in the domain of liquid-level sensing is yet to be explored completely. While inductive sensors are widely used as position and proximity sensors [17], the sensing mechanism to sense water level has not been clearly established and needs further research. In most of the existing inductive sensor-based designs, the inductive sensor is used as a secondary sensor, wherein the level is first converted into a displacement, which is subsequently sensed using inductive sensors [21,22]. A wireless inductive–capacitive approach has been utilized to measure levels from containers that have interiors that are inaccessible for wired sensor installation [23,24]. A method to implement point-level sensing using inductive sensors is presented in [25]. A similar approach using a linear array of inductive sensors is presented in [26]. A preliminary study on using planar coil-based inductive sensing techniques for level sensing is presented in [27]. The methods listed above do not sense both level and quality from the same sensing element. 

Of the various indicators of water quality, its electrical conductivity (EC) has emerged as one that is widely used [5]. Since the concentration of ions increases with the increase in impurities in water, the EC value increases as the quality deteriorates [6]. Conductivity measurement is generally performed at the laboratory by immersing the probes of the conductivity meter into the water sample [28].

Although separate sensors provide either of the parameters—level or conductivity information—it is desirable to have both these values provided by a single sensing element. A method to measure level and conductivity is discussed in [29]. Here, the coils are kept above the water level. The output is not linear and the sensor structure is not modular to extend the level-sensing range.

In this paper, we present a simple, scalable, planar-coil-based level-sensing mechanism that provides water level information and indication about the conductivity, and hence quality, using a single sensing element. Since both are derived from the same sensing element, it saves the designer one sensor, its associated cost, maintenance, etc. The operation of the sensor, sensing mechanism and measurement approach, prototype development and experimental results are presented below. 

## 2. Materials and Methods

Figure 1a depicts a conventional capacitance-type level sensor with two electrodes. As illustrated, distributed capacitance is present between the two electrodes throughout its length. In the absence of water, the total capacitance measured across the input terminal is determined by the dimensions of the electrodes and the dielectric properties of the strip (e.g., FRP) on which the electrodes are patterned. When water is present, as shown in Figure 1a, the capacitance of that area of the electrodes immersed in water alone changes (increases) as the relative permittivity of the water is close to 80 [30]. As relative permittivity comes in the numerator of the expression for capacitance, for parallel-plate and planar configurations [31], the value of capacitance measured across the terminals varies with respect to the level of water present in the storage unit or tank in which the sensor has been introduced. Although this is a well-accepted technique for intrusive level measurement, it has not been used to determine water quality. 

Conductivity of water has proven to be a good indicator of its quality [32]; e.g., total dissolved solids can be estimated from the conductivity of water. There are several methods available to measure the conductivity of water [32], and inductive sensors are one of the most widely used ones as they are quite rugged and do not require direct contact with the water, in contrast to the contact-based conductivity measurement schemes, which are prone to errors due to contamination of electrodes [28].

Figure 1b shows an easy-to-fabricate planar coil, with turn-to-turn parasitic capacitance. When the coil is excited using a time-varying voltage, e.g., a sinusoidal signal, the voltage level of individual turns is different as there is current through the parasitic capacitances. Thus, if we consider the input impedance of the coil across its terminals, there is a contribution due to the parasitic capacitance. If the coil is partially or completely immersed in water, the values of these parasitic capacitances increase as in the case of the capacitive level sensor illustrated in Figure 1a. At the same time, as the typical tap or drinking water is relatively conductive, there is an eddy current in the water. This is produced by the time-varying magnetic field of the coil when it is excited using a time-varying voltage. The magnetic field produced by the eddy current opposes the field caused by the measurement current in the coil, leading to a reduction in the flux linkage and, hence, reduction in the self-inductance value of the coil. If the water’s conductivity is higher, this effect is greater, leading to more reduction in the inductance of the coil. This is a function of the frequency of excitation, used to measure the inductance, as the skin depth of the water varies with conductivity. The planar coil has an insulation layer, which ensures no direct contact of the coil with water. There is parasitic capacitance that predominately comprises the capacitance formed due to the conductive coil surface, the insulation in front of it, and the water (relatively conductive) in immediate contact with it. As the water becomes more conductive, the effective capacitance changes (higher) when the coils see less conductive water. Thus, the coil’s impedance has information about the level at which water is present and its conductivity. The coil with self-inductance, parasitic capacitance, and winding resistance is represented by using the simplified (lumped) electrical equivalent circuit, which is shown in Figure 1c for further analysis. The input impedance of the coil between the terminals *p* and *q* is a function of the various parameters given in (1):(1)$$Z_{in}=f\left\{L_{c}\;\left[h,\;\sigma ,\;\delta \right],C_{w}\;\left[h,\;\varepsilon ,\;\sigma \right],\;R_{c}\;\left[T\right],\omega \right\}$$

In (1), $L_{c}$, $C_{w}$, and $R_{c}$ are the inductance, capacitance, and resistance of the coil, and $\omega =2\pi f_{ex}$ is the angular frequency of excitation, and $f_{ex}$ is the frequency of excitation. $L_{c}$ and $C_{w}$ are functions of level (*h*), conductivity (σ), and skin depth (δ) of water, as mentioned above. $R_{c}$ changes as a function of the temperature *T* of the water to which the coil is exposed.

In this work, we are making use of the change in capacitance between turn-to-turn of the coils and change in the self-inductance of the coil as a function of level of water and conductivity of water. The sensor’s design is modular; the coil is designed so that the range can be extended by adding similar coil units in series. A pictorial representation of the sensor in a tank is shown in Figure 2a. The coil is made of standard PCB process and is relatively less expensive. A photograph of one of the PCB units developed as a prototype is shown in Figure 2b, the dimensions of the coils in Figure 2c, and a zoom-in view of the fabricated PCB is shown in Figure 2d. 

It is a double layer PCB with two terminals *p* and *q*. Coils on the top and bottom sides of the PCB are connected in series, in a way that the magnetic fields generated by both sides aid. Capacitance between turn-to-turn is a function of the distance between the traces, the thickness of the insulation present in front of it, and the permittivity and level of the water column. 

A number of PCBs can be connected in series, as shown in Figure 2a, to extend the level-sensing range of the sensor. This series combination can be connected to a measurement system, where the coil’s impedance can be measured to provide information about the change in the coil’s inductance and the change in the coil’s parasitic capacitance.

### 2.1. Impedance of the Planar Coil

Input impedance of the planar coil with a capacitance in parallel, as shown in Figure 1d, can be obtained as in (2), which can be rewritten as in (3). The reactance part of (3) can be obtained as in (4).
(2)$$Z_{in}=\frac{R_{c}+j\omega L_{c}}{1-\omega^{2}L_{c}C_{w}+j\omega R_{c}C_{w}}$$
(3)$$Z_{in}=\frac{(R_{c}+j\omega L_{c})\left(1-\omega^{2}L_{c}C_{w}-j\omega R_{c}C_{w}\right)}{{\left(1-\omega^{2}L_{c}C_{w}\right)}^{2}+{\left(\omega R_{c}C_{w}\right)}^{2}}$$
(4)$$X_{in}=\frac{j\left(\omega L_{c}-\omega^{3}L_{c}^{2}C_{w}-\omega R_{c}^{2}C_{w}\right)}{{\left(1-\omega^{2}L_{c}C_{w}\right)}^{2}+{\left(\omega R_{c}C_{w}\right)}^{2}}$$

From (3), the parallel resonance frequency $f_{p}$ is written as in (5).
(5)$$f_{p}=\frac{1}{2\pi \;{\left(L_{c}C_{w}\right)}^{0.5}}$$

### 2.2. Experimental Setup 

To evaluate the practicality of the planar coil sensing element in sensing the water’s level and quality (conductivity), a prototype sensing coil was designed and fabricated, an experimental setup was formed, and tests were conducted. Details of the experimental setup are presented in Figure 3. The prototype PCB was introduced into a small tank for testing. The terminals of the PCBs were connected to a switch that can switch the coil between (i) a low-frequency impedance measurement unit, and (ii) an inductance-to-digital converter (LDC), LDC 1614 from Texas Instruments, which has 28-bit resolution. For (i), the impedance analyzer of ELVIS-II, from National Instruments was employed. As illustrated in the block diagram in Figure 3a, (i) and (ii) were connected to a computer. The computer has the tools to read from ELVIS-II and LDC 1614 and record the readings. A photograph of the test setup is given in Figure 3b, and the two PCBs fabricated for testing are shown in Figure 3c. As mentioned above, they can be mechanically joined using nuts and bolts while being electrically connected in series. For testing, only one coil was used. As the PCBs are identical, the series connection of sensing units is expected to have the same characteristic.

As shown in Figure 2c, the width of the PCB is 10 cm, and its height is 50 cm. There is a 2 cm border on the top and bottom, and a 1 cm border on both sides, leading to an effective coil height of 46 cm and width of 8 cm. In this area, 39 turns are accommodated on each side. As mentioned previously, the turns on both sides are in series. The inductance of the coil measured at 1 kHz is 1.530 mH. 

For frequency $f_{ex}$,$\;1\gg \omega^{2}L_{c}C_{w},\;$ which is much lower than $f_{p}$. Additionally, the coil resistance for the planar coils is usually in tens of ohms (for the prototype, it was 113 Ω), and $C_{w}$ is in pF or nF range depending on the coil size. For the prototype, $C_{w}$ = 203.8 pF was estimated based on the parallel resonance frequency observed as $f_{p}$ = 285 kHz. For these values, at 1 kHz, the denominator of (4) can be approximated as unity and reactance (Xin-LF) or ${X_{in}|}_{\left(f=1\;\mathrm{kHz}\right)}\;$ can be approximated as in (6).
(6)$${X_{in}|}_{\left(f=1\;\mathrm{kHz}\right)}≅j\omega L_{c}$$

## 3. Results

In the first experiment, the tank was filled with tap water in steps of 3 inches (7.62 cm) from an empty tank to its full level, which is 46 cm. The conductivity of the water was about 190 µS/cm. In each step, the reactance of the PCB was measured using the impedance analyzer of ELVIS-II, at 1 kHz. Simultaneously the output of the LDC 1614 was also recorded. The results are presented in Figure 4a,b, respectively.

Figure 4a shows that the reactance measured at 1 kHz reduces gradually as the level increases. Since the water has some conductivity (190 µS/cm, in this case), some eddy current is induced in the water in immediate contact with the PCB coil, which opposes the magnetic field by the coil, reducing its inductance. This effect increases as more of the coil is exposed to water, or the effective *L_c_* decreases due to the presence of the water. This is reflected in the reactance as per (6). 

The method used by the LDC 1614 to measure the inductance relies on the resonance frequency of the coil connected to it. It measures the parallel resonance frequency, assuming that the capacitance remains unchanged and there is change only in the inductance. Using this information, *L_c_* is calculated using (5), as all other parameters are known. However, in the case of the proposed sensing approach, as the coils are exposed to water, the capacitance changes depending on the fill level, i.e., increases with respect to an increase in fill level. This change was found to be more than the reduction in the *L_c_* owing to the eddy current in the water in contact with the coil. Compared to the low-frequency excitation, the skin depth is much less in the resonance frequency, which is 285 kHz in the case of the prototype. This reduces the eddy current drastically and hence the change in (reduction in) *L_c_*. Thus, here, the output of the LDC 1614 is a function of *L_c_C_w_*; as long as the total quantity increases (decreases) the output increases (decreases). This explains the reason for the increasing output of the LDC 1614 with level as observed in Figure 4b. Further experiments revealed that the LDC 1614 output has more sensitivity to the conductivity of water, causing the output to be nonlinear. Thus, it was decided to use the reactance measurement at 1 kHz for level indication.

### 3.1. Effect of Hysteresis

It has been observed from the initial studies that the reactance measurement results exhibit a hysteresis effect; the characteristic differed when we drained the water compared to filling it. We identified the source of this error as the presence of a water layer and droplets sticking to the PCB when we drain the water. To remove this error, we applied hydrophobic paint on all sides of the PCB. To confirm the improvement, we first painted half of the total height of the PCB, as shown in Figure 5, and repeated the filling and draining experiment. As expected, the effect of hysteresis was much less for the painted region. A sample result is available in Figure 5. Based on this study, the full surface of the PCB was painted with hydrophobic coating and the same was used for the subsequent studies. 

### 3.2. Repeatability of Level Measurement

To verify the repeatability of the level sensed by using the impedance measurement of planar coil at low frequency, several measurements were made using the hydrophobic coated PCB. The test tank was filled six times and drained, and measurements were recorded for every 3-inch step size. From these readings, the data with maximum deviations were noted and plotted in Figure 6. Figure 6 gives the average of these six measurements at each level. A linear fit, using the method of least squares, is made through the averaged data, which is given in Figure 6. The linear fit equation and R^2^ values are also presented in Figure 6. The deviation of maximum deviation (error-1), and second maximum deviation (error-2) from the linear-fit equation, respectively, at each level is also computed and the same is presented in Figure 6. It can be seen that the maximum deviation from the estimated level is 3 cm. Resolution of the measuring tape (0.25 inch = 0.635 cm) employed contributes to this deviation. The primary focus of this study was to observe the overall characteristic and its repeatability. The study confirms that the proposed level sensor is repeatable, and the level can be estimated with good accuracy by using the linear-fit equation. Accuracy can be improved if an instrument with higher accuracy is used for the impedance measurement at 1 kHz. 

### 3.3. Effect of Conductivity of Water 

For all the studies presented above, the same tap water with a conductivity of 190 µS/cm was used. Additionally, all the tests were done at a room temperature of about 35 degrees Celsius. In order to study the effect of conductivity on the reactance and LDC output, salt was added to the water and stirred properly. The resulting conductivity was more than 50 mS/cm. This salt water was used to fill the test tank and the measurements were taken: (i) reactance at 1 kHz and (ii) output of the LDC. The results obtained from (i) are presented in Figure 7. As can be seen, the reactance of the salt water at 1 kHz is much less than that of normal water, but the linearity of the output is retained even when the water is conductive. The reduction in sensitivity and offset is expected as the eddy current is higher in the salt water, as it has higher conductivity than normal tap water. Because of this, the inductance of the coil decreases. Thus, the results are as expected. However, this introduces a large error in the level estimation if we use the same linear-fit equation. To correct for this, a compensation method is needed. For this, we propose to introduce a small PCB at the bottom of the measurement PCB. It can be a square-shaped PCB with only a centimeter of width, and can be kept horizontal to the level-measuring PCB. Reactance of this new, small PCB is sensitive only to the conductivity and not to the level of the water, as it is kept horizontal to the main sensing PCB. Hence, the effect of conductivity on the level can be corrected if the ratio of reactance of the level-sensing PCB to that of the small reference PCB is taken. Such a correction is applied and the compensated results are shown in Figure 7.

The results obtained from the LDC for the salty water are presented in Figure 8. A huge relative change in output is noticeable when the water is conductive, and it is not linear. When the conductivity is more, there is a reduction in *L_c_*, as stated earlier. Thus, the output of LDC decreases with the increase in water conductivity. In Figure 4b, it can be observed that the output of the LDC increases with level. It has been understood that this effect occurs due to the increase in the capacitance of the planar coil due to the high permittivity (relative permittivity of 80) of the water. Additionally, at resonance frequency, inductive and capacitive reactances have equal value; furthermore, the change (increase) in capacitance is more than the decrease in *L_c_*. This explanation is not sufficient for the large change in the LDC output seen when the water is conductive. To understand this more, a part of the PCB with coil is closely examined; an expanded view is given in Figure 9. A cross-sectional view of the PCB and a close view of the PCB, indicating two copper traces, the insulation layer in front of it, and the water, are given in Figure 9. As indicated in Figure 9, there is capacitance *C_P_*_1_ and *C_P_*_2_ formed due to the PCB insulator layer, copper traces, and the water column present. Then, the water column present just in front of the PCB trace forms a parallel R–C network, where *R_wa_* represents the resistance of the current path through which a part of the current through *C_P_*_1_ and *C_P_*_2_ completes its path, and the rest of the current flows through *C_wa_*, which represents the capacitance provided by the water that is in parallel to *R_wa_*. 

When the conductivity is increased, *R_wa_* can become much lower than the reactance of *C_wa_*. In such a case, total impedance/reactance of the path *C_P_*_1_–*C_P_*_2_–*R_wa_* is decided by the series combination of *C_P_*_1_ and *C_P_*_2_ alone, which is lower than the combination *C_P_*_1_–*C_P_*_2_–*C_wa_*, or the equivalent capacitance value is much more than the value arrived at for clean water. Because of this, the *L_c_C_w_* increases noticeably and hence the reading of LDC increases significantly, compared to the one for clean water. As this change is significant, it is a very good indicator of conductivity and therefore of water quality. The exact value of conductivity can be obtained through calibration, but in this work, the objective it is to detect if the water is salty, and measure the level accurately. When a reactance measurement (compensated) is obtained from the sensor at low frequency (1 kHz for the prototype developed), the level is measured. For this level, there is an expected output from the LDC. If the LDC output is more than the expected value for a level given by the reactance measurement, it indicates that the quality of water was compromised. Currently, the quality indicator is serving as a detector; the value of conductivity and, hence, a water quality parameter such as TDS, if needed, can be obtained through calibration. 

## 4. Discussion

We compared the features of the proposed sensor with the existing ones. Table 1 presents the important sensing techniques available for level sensing and a comparison of those with respect to the proposed inductive–capacitive level sensor. The important limitations are also given in Table 1. The methods based on ultrasonic [7,33], image [8], lidar [12], and inductive [29] do not require direct contact with water, which is an advantage. Still, these methods cannot indicate change in water conductivity, except the one presented in [29]. For the technique presented in [29], however, the level-sensing range is not scalable, while the proposed one is modular in structure and range can be extended by adding additional modules. Further, [29] uses wound coils that are not easy to manufacture, while the proposed one uses planar coils made using standard PCB technology. The level sensor output is linear for the proposed one, while it is not for the case of [29]. 

While PCB coil manufacturing has its own advantages, due to the variations in the PCB manufacturing process, there is an associated variability in the resistance, inductance, and capacitance of the fabricated coils on the PCBs [35]. Among these, the resistance and capacitance have more variability due to the variations in the surface area, while inductance is expected to have less effect. To evaluate this effect, a sufficient number of PCB coils need to be made and tested. A calibration mechanism may have to be employed for each PCB coil set if there is noticeable variability. 

The correlation between the conductivity of water and quality, for example, total dissolved solids (TDS), is a well-studied topic [32]. We used this as a basis to add salt to change conductivity and test whether the sensor detects it or not. While the proposed senor is sensitive to conductivity, it is not sensitive to any specific microbes. A different sensing approach will be needed if the objective is to detect certain microbes. The PCB-based coils used for the study are insulated from water using hydrophobic paint. Hence, there is no direct contact present between the PCB and water. The long-term effect of such paint or the PCB material itself is not known to damage water quality, but a detailed study may be needed to confirm this. 

## 5. Conclusions

The feasibility of a planar coil-based sensor to measure water level, and at the same time detect the quality of the water, based on its conductivity, was conducted. The planar coil sensor was designed, fabricated on a PCB, and tested in a tank of clean water and salty water. The reactance of the coil measured at a very low frequency compared to its parallel resonance frequency is indicative of the tank’s water level. The sensing technique used in this case relies on the eddy current induced in the water exposed to the planar coil. Of course, this output is sensitive to the conductivity of water, but the linearity of the output with respect to level is maintained independent of the conductivity. Thus, the effect of conductivity on the output can be corrected by taking a ratio of the reactance of the sensing coil to that of a small reference coil exposed to the same water. 

Another measurement performed on the same planar coil sensing element, using an LDC, gave increasing output with level. It is concluded that this occurred due to the increase in the turn-to-turn capacitance in the planar coil when it is exposed to water, which has high relative permittivity. This measurement can also be used to sense level, but when the conductivity of water changes, the effective change in capacitance is larger compared to the change in inductance due to the eddy current effect. The LDC output was observed to have a noticeable change in the output from the expected reading for the same level but with higher salt levels, which indicates the relative increase in the conductivity of the water and, hence, reduced quality.

Thus, the proposed low-cost, low-power, easy-to-manufacture, modular level sensor with a water quality indication feature is suitable to realize cost-effective, large-scale smart water management systems, without compromising efficiency.

## Figures and Tables

**Figure 1 sensors-22-05508-f001:**
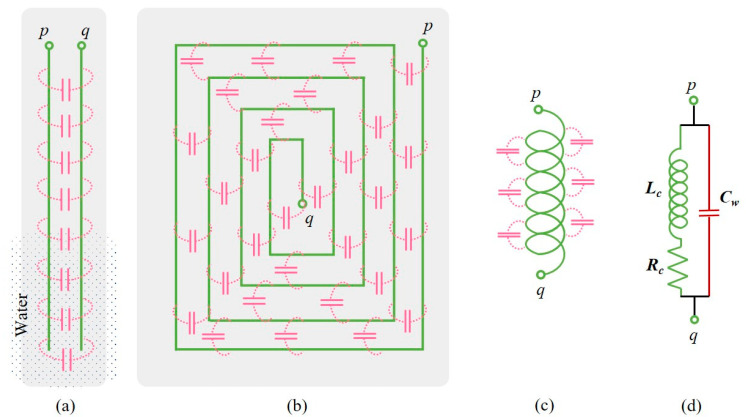
(**a**) Capacitive level sensor, (**b**–**d**) planar coil with parasitic capacitance and simplified electrical equivalent circuit.

**Figure 2 sensors-22-05508-f002:**
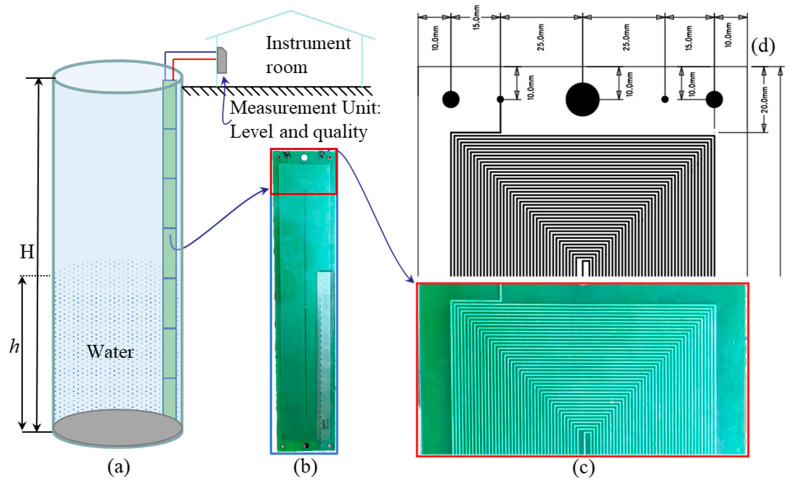
Proposed inductive–capacitive level sensor and the details of the fabricated sensor coil used for testing. It is a scalable, modular design; multiple units of PCBs with coil as shown can be connected in series and mechanically appended, as shown in (**a**), to extend the range of level to be sensed. (**b**) Prototype CB with coil, (**c**) close-in view of the PCB, and (**d**) important dimensions of the PCB.

**Figure 3 sensors-22-05508-f003:**
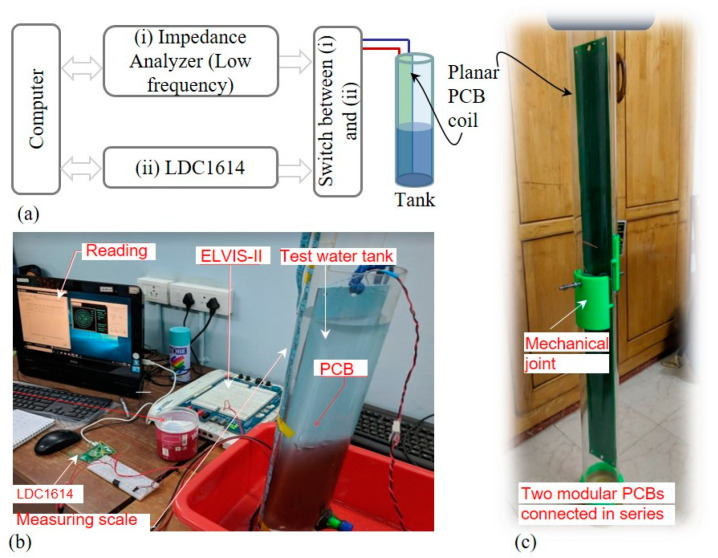
Experimental setup: (**a**) block diagram of the setup; (**b**) laboratory setup for testing; (**c**) prototype planar coils, connected electrically and mechanically in series to extend the range.

**Figure 4 sensors-22-05508-f004:**
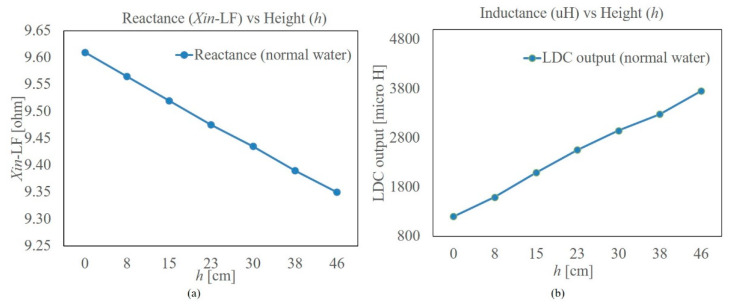
(**a**) Input reactance of the planar coil with respect to level (*h*) measured at 1 kHz. (**b**) Output from the LDC for the same water level.

**Figure 5 sensors-22-05508-f005:**
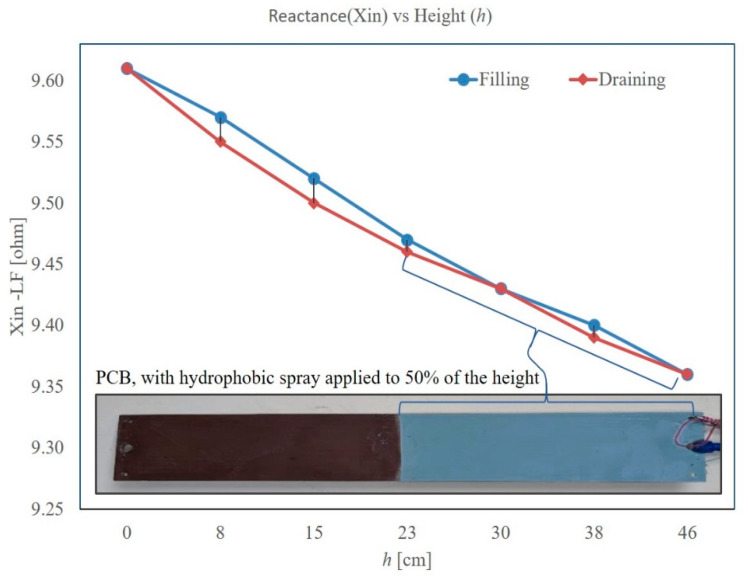
Hysteresis and effect of hydrophobic coating.

**Figure 6 sensors-22-05508-f006:**
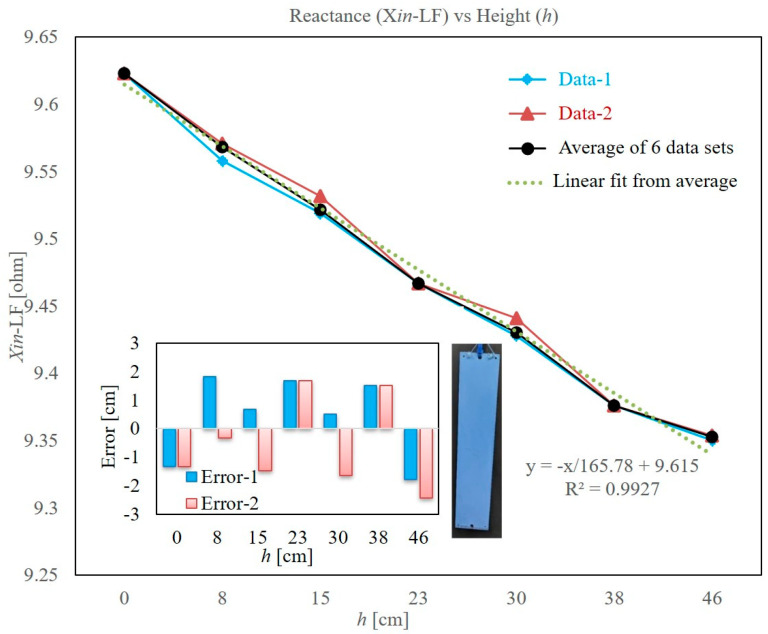
Results of repeatability test. The hydrophobic painted PCB used is shown in the inset photograph. Error in the estimated level with respect to the linear-fit equation is shown in the inset graph. Error-1 and error-2 represent two sets of data where the maximum and second maximum deviations are noted, respectively.

**Figure 7 sensors-22-05508-f007:**
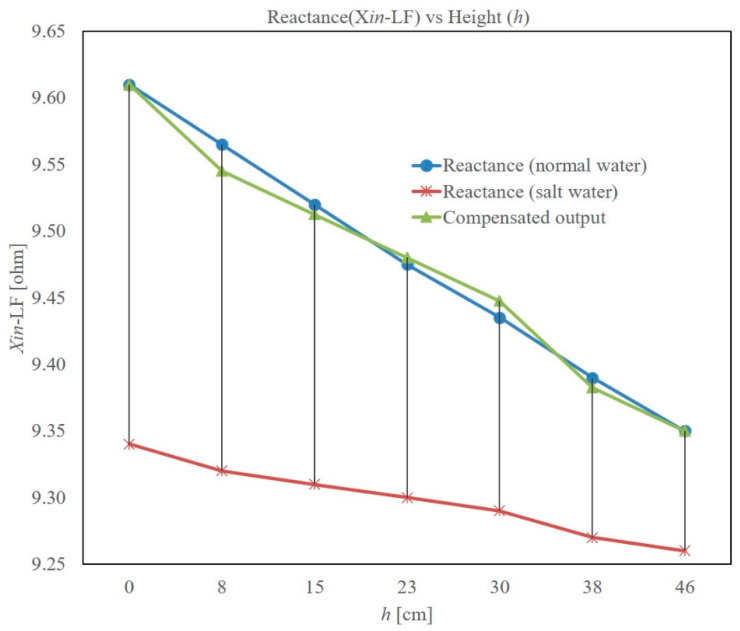
Effect of conductivity on the input impedance measured at 1 kHz, and compensated output.

**Figure 8 sensors-22-05508-f008:**
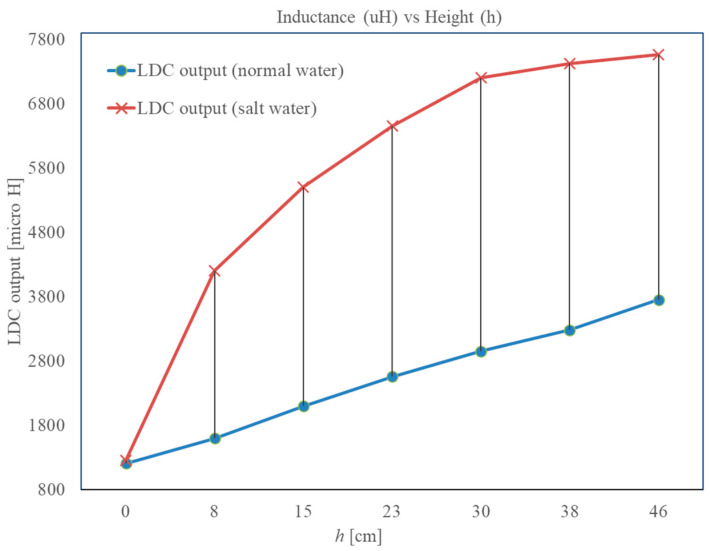
Output of the LDC for good water and salty/relatively conductive water.

**Figure 9 sensors-22-05508-f009:**
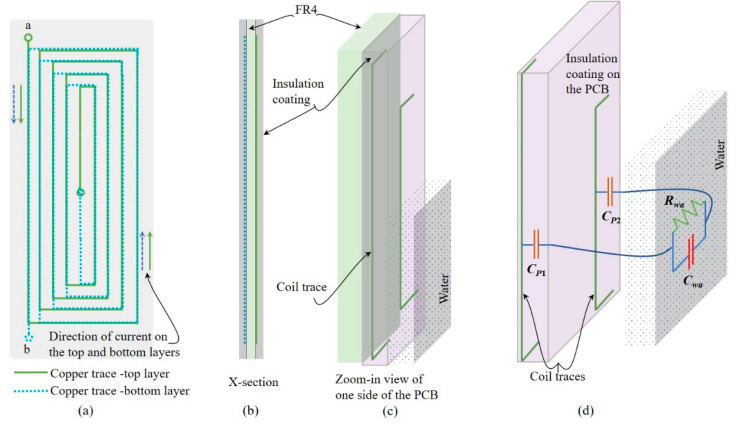
Details of the planar coil: (**a**) the direction of current in the coil; (**b**,**c**) views of the PCB. It is a two-layer PCB. The terminals of the coils are marked as ‘a’ and ‘b’. The traces are insulated and hence do not have direct contact to water. (**d**) Water column impedance and capacitance between the traces.

**Table 1 sensors-22-05508-t001:** Comparison with the important level-sensing methods.

Ref., Year	Sensing Technique	Easy to Manufacture	Detects Change in σ	Level Sensor			
				Range	Modular Range	Linear Output	Accuracy
[7], 2021 [33], 2019	Ultrasonic	Moderate	No	10 m	No	Yes	a few mm ^#^
[8], 2020	Image	Moderate	No	1.5 m	No	Yes *	1.5 cm
[9], 2015	Capacitive	Yes	No	1 m	Yes	Yes ^$^	0.8 cm
[10], 2020	Pressure	Moderate	No	0.5 m ^+^	No	Yes	a few mm ^+^
[12], 2020	Lidar	Moderate	No	10 m	No	Yes	1 cm ^#^
[34], 2022	Optical	Complex	No	0.8 m	Yes	Yes	1 mm
[24], 2020	Float, Inductive	Moderate	No	0.2 m	Yes	Yes	1%, 2 mm
[27], 2021	Inductive	Yes	No	0.15 m	No	NA	NA **
[29], 2008	Inductive	Moderate	Yes	0.3 m	No	No	2%, 0.6 cm
Proposed	Inductive–Capacitive	Yes	Yes	0.45 m	Yes	Yes	2.5 cm ^&^

^#^ Output is sensitive to temperature. Drift in the clock frequency of the electronics due to the temperature needs special attention. A maximum error of about 10 cm for 20 to 80 deg. Celsius change is reported in [12]. * Requires relatively complex signal processing. ^$^ Free water flow through the sensor tube is required, which may be affected due to scaling. ^+^ This sensor is tested for a 50 cm range. In general, level in tens of meters can be measured easily. The quantity estimation is sensitive to the density of water; accuracy is a function of it and accuracy of the pressure sensor. NA—not available. ** This is a preliminary study suggesting the possibility. ^&^ Accuracy of the impedance measurement system used in 2%. The reference level was measured using a measuring tape with 0.5 inch resolution. The focus of the work was to show the new sensing mechanism and functionality. A fine-tuned measurement system for this sensor is needed for field-level implementation.

## Data Availability

Not applicable.

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
