# Peer review of "Feasibility of a Planar Coil-Based Inductive-Capacitive Water Level Sensor with a Quality-Detection Feature: An Experimental Study"

_sensors, 2022, doi:10.3390/s22155508_

Round 1
Reviewer 1 Report
This manuscript presents a low-cost, low-power and scalable planar coil-based level sensing mechanism that measuring the water level and quality in a single sensing element. Overall, the manuscript is interesting and well organized. However, several parts were not clearly described. I would suggest a minor revision and the comments are list below.
1. There are some reported works that can simultaneously measure water level and conductivity, such as reference [29]. Authors should compare their work with already reported work. A brief table of comparison should be given in the revised manuscript.
2. There are too many figures in the manuscript. Some Figures need to be merged or put in SI.
3. The introduction part should be more succinct.
4. What's the device-to-device variability? What's the N of measurement and statistical analysis?
Author Response
We are very thankful to the Editor and the Reviewers for their valuable and helpful comments to improve the quality of the paper.
We have revised our manuscript along the lines suggested by the Editor and the Reviewers. Detailed answers to the questions posed by the Reviewers, descriptions of how we have incorporated the Reviewers’ comments in preparing our revised manuscript, and the revised content, highlighted in yellow, are given below.

Reviewer 2 Report
The paper discusses the potential of using the Planar Coil-Based Inductive-Capacitive methodology for measuring the water level and also the water quality at the same time.
However, I feel the paper lacks some aspects such as:
- How Novel is the proposed approaches compared with existing ones in the literature? if t outperforms them, then why not the authors provides a comparison with other approaches in terms of accuracy, pros and cons.
- About the fact how the proposed solution detect the water quality, the authors gave an example of adding salt to the water and how it affected the conductivities. How about the effects of microbes, germs, and other factors that can contaminate the water without affecting its conductivity? Finally, will the immersed coil itself affect the water quality? if no, how to you prove that?
I feel wireless based methods for measuring the water level is both safter and more convenient than these solutions that involve direct contact with water.
Author Response

(The authors gave the same response as above.)

Reviewer 3 Report
This work studied the quality water by measuring the conductivity from a planar coil-based sensor. It can be further improved by comparing the detected results (lowest resistance that could be detected) with some commercial sensors. Secondly, the figures need to be in a constant style (Figure 5 and 7). Third, some typos and grammar are found in line 378.
Author Response

(The authors gave the same response as above.)

Round 2
Reviewer 2 Report
The reviewers have addressed my comments.